Habitat suitability—density relationship in an endangered woodland species: the case of the Blue Chaffinch (Fringilla polatzeki)

Carrascal Luis M. lmcarrascal@mncn.csic.es 1
Moreno Ángel C. 2
Delgado Alejandro 3
Suárez Víctor 3
Trujillo Domingo 3
1 Department of Biogeography & Global Change, Museo Nacional de Ciencias Naturales, CSIC , Madrid , Spain
2 Viceconsejería de Medio Ambiente, Gobierno de Canarias, Dirección General de Protección de la Naturaleza , Las Palmas de Gran Canaria , Spain
3 Wildlife Freelance , Las Palmas de Gran Canaria , Spain
Pimm Stuart
Electronic publication date: 2017 Sep 12
Publication date: 2017
Volume: 5
Electronic Location ID: e3771
Received 2017 Apr 21; Accepted 2017 Aug 16
Copyright: ©2017 Carrascal et al.
Copyright year: 2017
Copyright holder: Carrascal et al.
License: This is an open access article distributed under the terms of the Creative Commons Attribution License, which permits unrestricted use, distribution, reproduction and adaptation in any medium and for any purpose provided that it is properly attributed. For attribution, the original author(s), title, publication source (PeerJ) and either DOI or URL of the article must be cited.
License URL: https://creativecommons.org/licenses/by/4.0/

Keywords: Abundance-suitability relationship, Blue chaffinch, Canary islands, Population size and density, Habitat suitability modelling

Funding: Conservation Program for the Blue Chaffinch Cabildo de Gran Canaria European Union LIFE94 NAT/E/001159 LIFE98 NAT/E/005354 LIFE14 NAT/ES/000077 All the funding and sources of support come from the Conservation Program for the Blue Chaffinch implemented by the Gobierno de Canarias throughout 1991–2004, Cabildo de Gran Canaria (2005–2015), and by European Union (1995–1996: LIFE94 NAT/E/ 001159, 1999–2002: LIFE98 NAT/E/005354, and 2016: LIFE14 NAT/ES/000077). There was no additional external funding received for this study. The funders had no role in study design, data collection and analysis, decision to publish, or preparation of the manuscript.

==============================
Background

Understanding constraints to the distribution of threatened species may help to ascertain whether there are other suitable sectors for reducing the risks associated with species that are recorded in only one protected locality, and to inform about the suitability of other areas for reintroduction or translocation programs.

Methods

We studied the Gran Canaria blue chaffinch (Fringilla polatzeki), a habitat specialist endemic of the Canary Islands restricted to the pine forest of Inagua, the only area where the species has been naturally present as a regular breeder in the last 25 years. A suitability distribution model using occurrences with demographic relevance (i.e., nest locations of successful breeding attempts analysed using boosted classification trees) was built considering orographic, climatic and habitat structure predictors. By means of a standardized survey program we monitored the yearly abundance of the species in 100 sectors since the declaration of Inagua as a Strict Nature Reserve in 1994.

Results

The variables with the highest relative importance in blue chaffinch habitat preferences were pine height, tree cover, altitude, and rainfall during the driest trimester (July–September). The observed local abundance of the blue chaffinch in Inagua (survey data) was significantly correlated with habitat suitability derived from modelling the location of successful nesting attempts (using linear and quantile regressions). The outcomes of the habitat suitability model were used to quantify the suitability of other natural, historic, pine forests of Gran Canaria. Tamadaba is the forest with most suitable woodland patches for the species. We estimated a population size of 195–430 blue chaffinches in Inagua since 2011 (95% CI), the smallest population size of a woodland passerine in the Western Palearctic.

Discussion

Habitat suitability obtained from modelling the location of successful breeding attempts is a good surrogate of the observed local abundance during the reproductive season. The outcomes of these models can be used for the identification of potential areas for the reintroduction of the species in other suitable pine forests and to inform forest management practices.

Introduction

Habitat suitability is usually determined by the relationship between environmental predictors and species occurrence or abundance (Acevedo et al., 2016). Using species occurrence to understand the suitability of habitat is commonly employed when studying very scarce and spatially restricted species. In the case of very mobile species, such as birds, the localities where they have been observed may include areas that are important for their existence (e.g., space around nesting places), as well as other marginal areas used while dispersing or foraging outside the core home range. Thus, the utility of species occurrence models rests on the availability of good data on local species distribution, which will be all the better as the localities are linked to processes directly related to survival or breeding success. On the other hand, analysis of the spatial variation of abundance may pose problems, since several authors have warned that density could be a misleading indicator of environmental quality if it is negatively correlated with other demographic variables via Ideal Pre-emptive Distribution processes (Van Horne, 1983; Pulliam & Danielson, 1991; Brawn & Robinson, 1996). For example, in environmentally restrictive areas, dominant individuals could displace other young or subordinate individuals to marginal areas where they become abundant, not as a consequence of habitat tracking considering foraging success, survival or successful reproduction, but according to mere habitat displacement. Therefore, in order to obtain good predictions about habitat suitability for selecting areas to protect remnant populations of endangered species, or for defining habitat for translocation, it is necessary to maximize data quality related to survival or breeding success. Furthermore, it is also necessary to know if habitat quality inferred from local abundance is associated with other independent measures related to suitability linked with demography (Vickery, Hunter & Wells, 1992). The “habitat suitability—abundance” equivalence is a subject of intensive research because independent tests are needed to ascertain the validity of predictions of species occurrence models, considering that presence data are much easier to obtain than local measures of density (Jiménez-Valverde, 2011; Weber et al., 2017).

Natural reserves are established to protect the whole biodiversity or those threatened species that have conservation problems (Geldmann et al., 2013). Nevertheless, their effectiveness may vary if phenomena outside the borders of the protected areas affect populations inside them (e.g., global warming and changes in rainfall regime, emergent diseases, invasive species), a worrying concern if species are restricted to only one protected area. This concern is a relevant question contributing to knowing whether it is advisable to place the emphasis on the conservation of an endangered species in only the protected area where it is relegated, or if more efforts should be directed towards translocations to other areas (Pérez et al., 2012; Rummel et al., 2016). To identify those other potential areas it is necessary to know constraints to the distribution of species restricted to only one protected area, in order to know if there are other suitable sectors for reducing the risks associated with the presence of an endangered species in only one locality (an IUCN criteria for cataloging threat; IUCN, 2012).

The blue chaffinch of the Gran Canaria island (Fringilla polatzeki, Canary Islands) is a recently established species on the basis of genetic, morphological and behavioural data (Pestano et al., 2000; Lifjeld et al., 2016; Sangster et al., 2016), mainly restricted to the Strict Nature Reserve of Inagua-Ojeda-Pajonales (Inagua, hereafter; 39.2 km2; Moreno & Rodríguez, 2007). It inhabits mature pine forests, where nests are placed in tall trees; breeding success is very low for a Fringillidae, with only ca. 1.5 fledglings per successful nesting attempt, and 1.4 clutches per breeding season (Rodríguez & Moreno, 2008; Delgado et al., 2016). The estimated population size of the Gran Canaria blue chaffinch (guessed at around 300 birds with no recent estimation in its whole area of distribution, BirdLife International, 2016a) lies within the left tail of the distribution of minimum viable population (MVP) estimates for many species, far away from the average MVP of 3,750 individuals for birds (Brook, Traill & Bradshaw, 2006; Traill, Bradshaw & Brook, 2007). This is most notable if we take into account the small size of the species (approx. 30 g), since body mass in birds is usually negatively correlated with abundance or maximum ecological densities in preferred habitats (Carrascal & Tellería, 1991; Gaston & Blackburn, 2000). Surprisingly, and in spite of its low population size and smaller distribution area in comparison with the also endemic blue chaffinch from Tenerife island (Fringilla teydea; Rodríguez & Moreno, 2004; Moreno & Rodríguez, 2007), it has a higher haplotype diversity of the mitochondrial DNA control region (Pestano et al., 2000).

The main goals of this study are twofold. Our first goal is to build a species occurrence distribution model considering orographic, climatic and habitat structure predictors. This goal is carried out relying on high-quality occurrence data, using the location of successful breeding attempts. The results of this model are used to contrast the habitat preferences of the Gran Canaria (F. polatzeki) and Tenerife (F. teydea) blue chaffinches considering the available literature, and to predict the habitat suitability of the natural and historic pine forests of Gran Canaria located within the same altitudinal range of Inagua. An applied utility of this aim is to understand if there are important environmental restrictions limiting the natural presence of the blue chaffinch outside of Inagua, and to quantify the suitability of other historic pine forests on Gran Canaria as candidates for future translocations of birds. Our second goal is to test if habitat suitability modelling, considering the location of successful nesting attempts, is related to independent measures of bird abundance during the breeding season using a different methodological approach. This exercise would cast light on the usefulness of occurrence distribution models, using labour-intensive occurrences with demographic relevance, forecasting the spatial variation of habitat suitability, and the validity of survey programs to derive estimates of environmental quality.

Material and Methods

Study areas and environmental data

The study areas are located in several pine forests of Gran Canaria (27°58′N, 15°35′W), an island of volcanic origin (1,560 km2, maximum altitude of 1,950 m.a.s.l.; for more details on the vegetation of the island see Santos, 2000). The canary pine forests are dry and monospecific stands of Pinus canariensis, very heterogeneous regarding the size and cover of trees and undergrowth (mainly composed by Leguminosae shrubs Adenocarpus spp. and Chamaecytisus proliferus, and the Ericaceae shrubs Erica arborea and E. scoparia), occupying semi-arid hilly terrains comprised of a predominance of high slopes and rugged terrain (González, Rodrigo & Suárez, 1986).

Figure 1 Study areas (green) in Gran Canaria island (Spain).

Other pine forests, outside the altitudinal range of the core distribution area of the blue chaffinch in Inagua, are also shown (Moriscos and La Cumbre; they are pine plantations mainly established after 1960). White dots in Inagua Natural Reserve show the location of nests with successful breeding attempts (at least one chick fledged, and only one nest per breeding pair and year). Black dots show the centre of 100 units of 229 m in length of a survey trail of 22.9 km repeated from 1994 to 2016.

The main study area is located in the pine forest of Inagua Integral Natural Reserve (37.59 km2 with nearby pine stands; Special Protection Area of the European Union since 1979), which harbours the main extant breeding population of the blue chaffinch (Moreno & Rodríguez, 2007). Location of nests and yearly monitoring of blue chaffinch abundance were carried out in Inagua. For evaluating the habitat suitability of other mature pine forests within the environmental span of Inagua, we also considered the pine forests of Tamadaba (28.12 km2), Pilancones (31.67 km2) and Tauro (4.70 km2). Figure 1, Table S1 and Figs. S1–S3 show the geographical location of the study areas and their environmental characteristics. The four pine forests show a broad overlap in orographic attributes, with all cardinal orientations represented: altitudinal range of the studied pine forests is 250–1,550 m a.s.l., slopes of the terrain varies between 0% and 260% (with very steep averages of 45%–55%). Pine canopy cover ranges between 0% (clearings) and 99%, with Tamadaba forest being the area with the largest cover (43%). Pine height also shows a large overlap among the four pine forests, with the tallest pines reaching 40 m in Inagua. The shrub layer shows similar structural characteristics in the four pine forests, with average covers ca. 10% (maximum of 75%) and heights ca. 0.7 m (maximum values of 1.25 m). Climatic variables considerably overlap among the study areas, with high levels of average incident sun radiation during April-August (ca. 7,000 kWh/m2; minimum of 4,567 and maximum of 7,515), high average temperatures in May (ca. 19 °C; minimum: 17.0 °C; maximum: 21.2 °C) and July (ca. 24.5 °C; minimum: 23.6 °C; maximum: 25.9 °C), and low summer rainfall (July–September) ranging from 0 mm to 34 mm (Tamadaba is the pine forest with the highest rainfall, mainly horizontal precipitation, while Pilancones is the driest pine forest).

A severe fire occurring in July 2007 badly affected the Inagua Reserve, Pilancones and Tauro, but not the Tamadaba forest (see Fig. 1 in Suárez et al., 2012). The Canary Pine has the remarkable characteristic of being able to survive and grow after fire. In most places the pine foliage was partially recovered by June 2008, and the tree foliage showed full growth by the breeding season of 2010.

The geographic information was managed using the GRASS 6.4 (GRASS Development Team, 2015). The cartographic information employed to generate the digital terrain model comes from the “Infraestructura de Datos Espaciales de Canarias” (http://www.idecanarias.es/). The digital elevation model was built from a contour map with 5-m equidistant topographic curves which was converted to a raster map of 50 × 50 m resolution, with module {v.to.rast} and {r.surf.contour}. From the digital terrain model, raster maps of slopes of the terrain, and cardinal orientations of the hillsides, were elaborated at 50 * 50 m resolution by means of the module {r.slope.aspect}. Climatic variables were obtained from the “Clima-Impacto” project (http://climaimpacto.eu/), developed by the Gobierno de Canarias and funded by the European Regional Development Fund of the European Union, at a raster resolution of 50 * 50 m. Vegetation structure variables (pine and shrubs covers and heights) were obtained from precision laser LiDAR measurements. Data was provided at a raster resolution of 25 * 25 m by the project “Enriquecimiento de la Cartografía de las islas forestales de Canarias a partir de datos LIDAR” (GESFORMAC -Gestión y Planificación Forestal en la Macaronesia-, funded by European Regional Development Fund and by Dirección General de Protección de la Naturaleza del Gobierno de Canarias). These vegetation LiDAR measurements were upscaled to a resolution of 50 * 50 mm using the module {r.resample}. Finally, solar radiation data were obtained from the photovoltaic potential maps in the Canary Islands (http://www.idecanarias.es/), partially funded by the Spanish Ministry of Industry, Tourism and Commerce, and by the European Regional Development Fund.

Bird survey and nest location of blue chaffinches

Data on bird counts was obtained from line-transect sampling in Inagua during the breeding season (second fortnight of May and the first fortnight of June; see Rodríguez & Moreno, 2008) from 1994 to 2016 in 15 different years. A fixed network of trails of a total length of 22.9 km has been surveyed using the same methodology since 1994 (see Fig. 1). From 1994 to 2006, a transect of 22.9 km was surveyed one time per year; from 2011 to 2016, the transect was repeated three times on different days to obtain more reliable results (i.e., using the average of the three surveys). Transects were carried out on windless and rainless days, walking along single tracks at a low speed (1–3 km/h approximately), during the first four hours after dawn. Different persons carried out the surveys: A.C.M. from 1994 to 2004; V.S and A.D, in 2006, 2011–2016. To account for inter-personal and between-year variation in detectability while collecting counts, we employed distance sampling methods (Buckland et al., 2007). For each bird heard or seen, the perpendicular distance to the observer’s trajectory was estimated. Previous training helped to reduce inter-observer variability in distance estimates. Detection distances were right-truncated, excluding 5% of birds recorded far away (i.e., beyond 125 m). The total length of transects were divided in 100 contiguous units of equal length (229 m), to which the detected blue chaffinches were averaged across years, accounting for detection probability.

Intensive surveys of the Inagua pine forest during 2011 to 2016 allowed the location of active nests (carried out by V.S., A.D. and D.T.). We restricted sampled nests used in analyses to those years when the pine forest had recovered after the forest fire of July 2007. Although searches were mainly carried out around the area covered by the fixed network of trails where the monitoring program was conducted, other sectors covering the whole Inagua reserve were surveyed while moving around to access those trails (by foot and by vehicle on dirt tracks). Nests were located by following individuals during the prelaying and incubation period (mainly by females), by means of audible begging calls by nestlings, or by observing parents feeding bouts to chicks (see Rodríguez & Moreno, 2008 for more details on nest location and the breeding biology of the blue chaffinch in Inagua). Nests were monitored every 3–5 days in order to establish the successful reproduction of each breeding pair. We considered a successful breeding attempt when at least one fledgling was produced in the focal nest. Fifty-nine successful nests were recorded: 16 in 2011, 12 in 2013, 16 in 2014, 15 in 2016 (Fig. 1). They were found within an area of 24.2 km2 (2.6 * 9.2 km in latitude and longitude geographical dimensions). Altitudinal range of nest locations was 860–1,485 m a.s.l., within a broad spectrum of orographic conditions regarding the cardinal orientation and the slope of the terrain (see Table S1). The Consejería de Medio Ambiente del Cabildo de Gran Canaria gave access to carry out all the field work under the LIFE14 NAT/ES/000077.

Data analyses

Detectability models for the blue chaffinch were built with the R packages {Distance} (Miller, 2016a) and {mrds} (Miller, 2016b) under R version 3.1.2 (R Core Team, 2014). Population density of the blue chaffinch in Inagua was calculated considering the counts of birds in the 22.9 km transect and the effective strip width (ESW) derived from the probability of detection.

Breeding habitat suitability for the blue chaffinch in Inagua was modelled using boosting classification trees (BCT) with the occurrence of the species denoted as nest locations where successful breeding occurred. Boosting trees are a statistical learning method that attains both accurate predictions and good explanations for regression and classification problems, dealing with many types of response and predictor variables (numeric or categorical) and loss functions (Gaussian, binomial, Poisson), and managing parsimoniously complex interactions among predictors (De’Ath, 2007; Elith, Leathwick & Hastie, 2008). Boosting trees algorithm aims to improve model accuracy by fitting several trees in a stage-wise process in which the first tree focuses on the raw data, the second tree on the residuals from the first tree, and so on. Final predictions are made through model averaging.

BCT models were built and summarized using the R packages {gbm} (Ridgeway, 2016), {dismo} (Hijmans et al., 2016), {ROCR} (Sing et al., 2015) and {psych} (Revelle, 2016). Model parameters were: bag fraction of 2/3, learning rate of 0.001, tree complexity of 5 (a maximum model complexity of 11 nodes-leaves and five splitting criteria), and minimum of 5 sampling units per inner node. We used a ten-fold approach in order to test the accuracy of predictions of BCT models. The discrimination ability of BCT models was estimated through the area under the curve (AUC) of the receiver operating characteristic (ROC) plot of sensitivity against 1-specificity.

The environmental characteristics of the cells of 50 * 50 m in which the successful nests were located (n = 59; “breeding success”, level 1 of a binomial distribution) were compared with those measured in an identical number of 50 * 50 m cells randomly obtained from the background of Inagua (59 out of 15,037 cells obtained by means of resampling without replacement; “available habitat”, level 0 of a binomial distribution; see predictor variables in Table S1). Moreover, to obtain a more robust approximation to the habitat occupancy during reproduction, bootstrapped samples of the fifty-nine 50 * 50 m cells with successful breeding were obtained (i.e., resampling with replacement to avoid outliers). This analytical approach is associated with the classic, and well-established, study of habitat selection in which habitat use is compared against habitat availability (Cody, 1985; Wiens, 1989), in such a way that the sample size of the availability records is determined by the sample size recorded for the individuals under study. Moreover, this approach shows good statistical properties in comparison with other presence-only analyses (Barbet-Massin et al., 2012; see also Warton & Aarts, 2013). BCT predictions (p) around 1 denote that the 50 * 50 m cells have environmental characteristics very similar to those shown by the nest locations with blue finch successful reproduction. Conversely, BCT predictions around 0 are related to 50 * 50 m cells with extremely different environmental characteristics for the successful reproduction of the species. Finally, when p = 0.5, the environmental characteristics of the 50 * 50 m cells are similar to the average of the habitat use and habitat availability samples.

We repeated the BCT models 20 times, using different bootstrap samples of the 50 × 50 m cells characterizing the habitat of the 59 breeding successful nests, and different random samples of 59 background cells of 50 * 50 m. The values obtained with these 20 models were averaged (accuracy parameters, relative importance of the 12 predictor variables, partial effects of each variable, and predictions for all 50 * 50 cells in Inagua, Tamadaba, Pilancones and Tauro).

BCT predictions of habitat suitability for the successful breeding of the blue chaffinch in the one-hundred 229-m transect units, of the abundance monitoring transect, were obtained by averaging the nearest sixteen 50 * 50 m cells (estimated by means of the Euclidean distance). Habitat suitability in these 100 transect units were regressed upon the average number of blue chaffinch counted in those years when the pine forest was not affected by the forest fire of July 2007 (i.e., 1994–2006 and 2011–2016; 15 years considered). The spatial eigenvector mapping analysis (SEVM) was carried out to account for spatial autocorrelation in the 100 transect units (Diniz-Filho & Bini, 2005; Dorman et al., 2007). SEVM is based on the idea that spatial arrangement of sample locations can be translated into explanatory variables that capture spatial effects, by means of the eigenfunction decomposition of the spatial connectivity matrix among the 100 transect units of 229 m. SEVM produced three spatial filters that reduced the spatial autocorrelation in the residuals of the regression model of chaffinch abundance on predicted habitat suitability for successful breeding (i.e., the residuals showed nonsignificant figures of spatial autocorrelation according to Moran’s I). SEVM was carried out using SAM package (v. 4.0; Rangel, Diniz-Filho & Bini, 2010). Due to deviations from homoscedasticity of the residuals across the predictions of the SEVM model, we used the heteroscedasticity-corrected coefficient covariance matrix to obtain the proper significance of habitat suitability and the three spatial filters (Zeileis, 2004); the HC4m estimator suggested by Cribari-Neto (2004) was used to further improve the performance in significance estimations, especially in the presence of influential observations under small sample sizes (using the R package {sandwich}, Lumley & Zeileiss, 2015). Quantile regression of bird abundance against habitat suitability was carried out using {quantreg} package (Koenker, 2016), applying the bootstrapping approach for estimating standard errors and significance.

The probable population size of the blue chaffinch in Inagua was estimated considering the suitability predictions in cells of 200 * 200 m2 (joining sixteen 50 * 50 m2 cells), the relationship between habitat suitability and local abundance of the blue chaffinch in 2011–2016 (see above; i.e., the equation converting the probability of occurrence in bird numbers), and the detectability in the period 2011–2016 (number of blue chaffinches corrected for detectability bias = predicted number of chaffinches divided by 0.56). We calculated the probable population size of the species in Inagua adding up the predictions of bird numbers in the sample of 200 * 200 m2 cells. The 95% confidence interval of the predictions was estimated by means of percentiles (i.e., 2.5% and 97.5%), after bootstrapping the probable number of birds in the 200 * 200 m2 cells (1,000 bootstraps).

Results

Breeding habitat selection and habitat suitability modelling

The boosted classification tree models (BCT) produced highly accurate results, considering sensitivity (0.999), specificity (0.979), 10-fold cross-validation AUC (0.905), and positive (0.979) and negative (0.999) predictive success figures (see Table 1 for more details regarding the results of the 20 randomized runs of the BCT models, each time with a different random sample of background 50 * 50 m cells). The variables with the highest relative importance in the BCT models were pine height (relative importance adding up to 100% = 26.4), tree cover (19.2), altitude (13.7), and rainfall during the driest trimester (July–September; 11.7). The remaining eight predictors had relative importance lower than that expected considering the number of predictors (100/12 = 8.3). Table 2 shows the results for the relative importance of predictors in 20 runs of the BCT models, and Fig. 2 shows the partial dependence plots for the four most influential variables.

Table 1 Summary of the 20 randomized runs of the boosted classification tree (BCT) models analysing habitat suitability of the nesting location of successful breeding pairs (at least one fledgling per season).

The BCT models compare the habitat characteristics in pixels of 50 × 50 m around nests (59 nests with breeding success recorded in six years from 2011 to 2016) against the same number of pixels of the same size randomly obtained from the pine forests of Inagua reserve. Twelve environmental variables were used in all BCT models (see Table 2).

	Mean	sd	Minimum	Maximum	
Number of boosted trees	4,640	996.1	2,800	6,400	
Ten-fold cross-validation AUC	0.905	0.024	0.869	0.938	
Sensitivity	0.999	0.004	0.983	1.000	
Specificity	0.979	0.015	0.932	1.000	
Negative predictive value	0.999	0.004	0.983	1.000	
Positive predictive value	0.979	0.015	0.937	1.000	

Table 2 Average relative importance (in %) of the 12 environmental variables used in boosted classification trees models (for more details see Table 1).

Results are for 20 randomized runs analysing habitat suitability of the nesting location of successful breeding pairs against the same number of pixels of the same size randomly obtained from the pine forests of Inagua reserve.

	Mean	sd	Minimum	Maximum	
Average pine height	26.4	9	10.2	49.2	
Cover of the canopy (pine) layer	19.2	8	7.7	37.5	
Altitude	13.7	6	2.2	24.3	
Rainfall in July–September	11.7	6	3.7	25.6	
Slope	5.6	3	2.8	9.6	
Incident solar radiation	5.0	3	1.7	14.5	
Northern orientation	4.4	2	2.1	7.9	
Cover of the shrub layer	4.3	1	1.5	7.4	
Average temperature in May	2.9	1	1.5	4.6	
Western orientation	2.6	1	1.6	4.8	
Average height of shrubs	2.2	1	0.5	5.3	
Average temperature in July	2.0	1	1.3	3.4	

Figure 2 Average partial dependence plots for the four most influential variables in the 20 randomized runs of boosted classification trees models analysing habitat suitability of the nesting location of successful breeding pairs of blue chaffinches against the same number of pixels of the same size randomly obtained from the pine forests of Inagua reserve.

Suitability value of 0.5 denotes random distribution according to each predictor (depicted by means of a dashed line). Values of the predictors with low suitability figures show that those environmental conditions are not favourable for the breeding success of the blue chaffinch in Inagua reserve. See Tables 1 and 2 for more details.

Habitat suitability for successful breeding steadily increased with pine height from 15 to 20 m (remaining stably high above the second value), with tree cover from 25% to 37% (the partial influence of tree cover was at random when cover was higher than 55%), with altitude from 1,100 to 1,280 m a.s.l. (remaining stably high above the second value), and from 13 to 20 mm of summer rainfall. Habitat suitability in Inagua was very low in sectors with <17 m of pine height, <30% of tree cover, at altitudes <1,100 m a.s.l. and at locations with <13 mm of precipitation during July–September. Mean habitat suitability in the forest patches with those characteristics was very low (0.029, sd = 0.019, interquartile range: 0.018–0.030, n = 2,285 cells). Conversely, habitat suitability reached the highest figures in woodland sectors located between 1,200 and 1,550 m of altitude, with pines taller than 20 m covering 37–50% of the area, and with a summer precipitation of 18–24 mm. Average habitat suitability in these favourable forest patches was very high (0.827, sd = 0.083, interquartile range: 0.781–0.889, n = 261 cells).

The average BCT model obtained in Inagua has been applied to the environmental data of the pine forests of Gran Canaria island located within the altitudinal range of the study area in which the BCT models were built. The results of the predicted suitability for the pine forests of Inagua, Tamadaba, Pilancones and Tauro are presented in Fig. 3, and with more detail in the Figs. S1–S3. Habitat suitabilities of pine forests are summarized in Fig. 4 according to the area in an increasing scale of suitability levels. Inagua is the pine forest with the largest surface for the successful breeding of the blue chaffinch (7.95 km2 with a suitability > 0.5), followed by Tamadaba pine forest (3.89 km2) and Pilancones (0.42 km2); Tauro forest lacks suitable habitat for the reproduction of the species. This pattern of among forests differences in habitat suitability becomes more skewed when considering higher levels of habitat suitability; e.g., with suitability >0.8, there are 2.09 km2 in Inagua, 0.48 km2 in Tamadaba and a complete lack of habitat in Pilancones and Tauro. Moreover, there is more contiguity of woodland patches with high levels of habitat suitability, and their sizes are larger, in Inagua than in Tamadaba (compare smoothed values of suitability >0.5 in Figs. S1 and S2). Finally, the proportion of pine forest surface with very low habitat suitability (e.g., <0.2) decreased according to the following order: Pilancones (92.1%), Tauro (89.2%), Tamadaba (62.5%) and Inagua (57.5%). Summarizing, Inagua reserve, the classical pine forest with historic and continuous presence of the blue chaffinch, has the largest potential area of the most favourable habitat for the successful breeding of the species, with larger and less fragmented suitable woodland patches, and with the lowest proportion of unfavourable breeding habitat. The pine forest of Tamadaba, with scarce presence of the blue chaffinch in the last 60 years, also provides suitable woodland patches for the species, although the amount of highly favourable habitat is lower, and its patchiness higher, than that obtained for Inagua. The pine forests of Pilancones and Tauro have an extremely low habitat suitability for the successful breeding of the species.

Figure 3 Habitat suitability map for the successful breeding of the blue chaffinch in four pine forests of Gran Canaria island located within the altitudinal range of Inagua.

The map resolution is 50 * 50 m2 cells. (A) Tamadaba, (B) Inagua, (C) Pilancones and Tauro. Black dots show the centre of 100 units of 229 m in length of a survey trail of 22.9 km repeated from 1994 to 2016.

Figure 4 Surface (in hectares, ha) of four pine forests of Gran Canaria Island with different levels of habitat suitability for the successful breeding of the blue chaffinch.

Relationship between local abundance and predicted habitat suitability

There was a positive relationship between the predicted breeding habitat suitability of BCT models in 100 units of the same 22.9 km survey trail in Inagua reserve, and the mean number of blue chaffinches counted in the breeding season during 15 years in those units (1994–2006 and 2011–2016, considering those years when the pine forest was not affected by the devastating forest fire of July 2007; Fig. 5). The linear model obtained taking into account three spatial autocorrelation filters (that reduced the spatial autocorrelation in the residuals of the model according to Moran’s I) was highly significant: R2 = 42.5%, F4,95 = 17.55, p < 0.001. The partial contribution of the spatial filters (i.e., spatial component) to total variance in blue chaffinch counts was 19.2%, that attributable to predicted suitability was 15.3%, while 8% was the shared contribution of both sets of predictors. The partial effect of the habitat suitability on finch counts was highly significant (partial slope = 0.661, heteroskedastic-corrected standard error = 0.151, p < 0.001). This relationship depicts an increasing error variance. In fact, a quantile regression analysis shows that the slope progressively increases from 10% to 50% to 90% percentiles (tau = 0.1, b = 0.367, se = 0.187, p = 0.053; tau = 0.5, b = 0.491, se = 0.214, p = 0.0243; tau = 0.9, b = 0.760, se = 0.251, p = 0.003; taking into account the three spatial autocorrelation filters). Thus, two different sets of habitat preference measures were highly correlated, showing that for a passerine species with a low population density, such as the blue chaffinch in Gran Canaria, local estimations of abundance are positively related to habitat suitability for successful breeding.

Figure 5 Relationship between the predicted breeding habitat suitability of BCT models and the average number of blue chaffinches counted during the breeding season in 100 transect units of 229 m along the same 22.9 km survey trail in Inagua reserve during 15 years (1994–2006 and 2011–2016 in those years when the pine forest was not affected by the devastating forest fire of July 2007).

The thick line shows the partial OLS regression slope, and the three dashed lines the regression slopes for 90%, 50% and 10% quantile regressions, after controlling by three spatial filters obtained by means of spatial eigenvector mapping (i.e., the residuals of models do not manifest statistically significant spatial autocorrelation according to Moran’s I).

Detectability estimations were as follow; Years 1994–2004: probability of detection (pDET) = 0.64, se = 0.12, sample size (N) = 345 bird contacts; Years 2006, 2011–2016: pDET = 0.56, se = 0.09, N = 385. Considering the suitability map of Fig. 3 (joining sixteen 50 * 50 m2 cells into 200 * 200 m2 cells), the relationship between habitat suitability and local abundance of the blue chaffinch in 2011–2016 (very similar to that depicted in Fig. 5; partial slope = 0.780, heteroskedastic-corrected standard error = 0.194, p = 0.001), and the detectability in the period 2011–2016 (probability of detection = 0.56), we calculated the probable population size of the species in Inagua. The mean estimate is 279 birds, with a 95% confidence interval of 195–430 chaffinches.

Discussion

Relationship between local abundance and predicted habitat suitability

Studies aimed at predicting species abundance from species occurrence distribution models have yielded mixed results (e.g., Conlisk et al., 2009; Jiménez-Valverde et al., 2009; Yañez Arenas et al., 2014; Carrascal et al., 2015; Basile et al., 2016). A recent meta-analysis (Weber et al., 2017) concluded that occurrence data can be a reasonable proxy for abundance, especially if local environmental variables are considered when dealing with the abundance-suitability relationship. Our results show that the observed local abundance of the blue chaffinch in Inagua (survey data) correlated with habitat suitability derived from modelling the location of successful breeding attempts. The relationship was relatively triangular (Fig. 5), denoting the asymmetric relationship between these two parameters: unsuitable woodland sectors can only have low blue chaffinch abundances, whereas very favorable sites can have high or low abundances (see VanDerWal et al., 2009; Jiménez-Valverde, 2011). This suggests the existence of other important factors responsible for the emergence of the triangular positive relationship, such as the “unsaturation” of the available habitat (i.e., there are not enough blue chaffinches to occupy the favorable woodland patches) or other unmodelled habitat features. For example, García-del Rey et al. (2009) and García-del Rey, Otto & Fernández-Palacios (2010) have shown the importance of structure and species identity of the shrub layer during the breeding season, as well as pine seed availability on the ground for feeding habitat selection during winter in F. teydea of Tenerife island. On the other hand, survey counts at very small spatial scales may be accounting for the mere presence of floaters or breeders outside the core area of the nesting place, as chaffinches (especially males) spend a considerable amount of time outside the breeding territories (e.g., Hanski & Haila, 1988 with Fringilla coelebs). Conservation biologists are warned to be cautious when relying on abundance estimations as surrogates of habitat quality (Van Horne, 1983), which is more accurately described with labor-intensive demographic research (Johnson, 2007). Nevertheless, our results suggest that local abundance is a good surrogate of environmental quality for successful nesting in the blue chaffinch, which agrees with other previous studies showing that birds are usually more abundant in habitats where per capita reproduction is highest (e.g., review by Bock & Zach, 2004 and Carrascal & Seoane, 2009).

Population size

In spite of the imperfect fit between habitat suitability for successful nesting and local bird abundance, regional abundance can be accurately predicted in an unbiased way from occurrence distribution models by the aggregation of local predictions, whose overpredictions and underpredictions can be counteracted (see Carrascal et al., 2015 for 21 terrestrial bird species in La Palma, Canary islands). Thus, the species occurrence distribution models can be used as a cost-effective tool to provide tentative population estimations when data from exhaustive census programs are not available. We estimated an exiguous population size of ca. 280 blue chaffinches in Inagua, which is consistent with its low population density and the small area of this pine forest (37.6 km2). Although the topic merits an exhaustive census program, this assessment should be considered as a first approximation to the population estimation in Inagua. Another 38 blue chaffinches can be added to those low numbers (minimum estimation; Rodríguez, 2016), given the recently established small population located at higher altitudes in La Cumbre (20.7 km2; from a captive breeding and translocation program; Delgado et al., 2016; Rodríguez, 2016). Therefore, with ∼320 individuals in 58.3 km2 of pine forests during the breeding season, the Gran Canaria blue chaffinch is the passerine with the lowest population size in the Western Palearctic (average density: 5.5 birds/km2). This population size is several times lower than that recorded for the other three specialists species of marginal woodlands with very small populations: Sitta whiteheadi (5,500 individuals in ca. 185 km2, 29.7 birds/km2; BirdLife International, 2016b), Phyrrula murina (1,000 individuals in ca. 100 km2, 10.0 birds/km2; BirdLife International, 2016c), and Sitta ledanti (350–1,500 individuals in ca. 700 km2, 0.5–2.0 birds/km2; BirdLife International, 2016d). Although the population size of the blue chaffinch is considerably lower than minimum viable population sizes suggested for birds (around 3,500 individuals for a persistence probability of 99% in 40 generations; Brook, Traill & Bradshaw, 2006; Traill, Bradshaw & Brook, 2007), its persistence with relatively constant numbers in Inagua during the last 20 years probably shows its high resilience against demographic risk factors. Nevertheless, it could very well be that the stochastic variability of the environment has been rather benign during the last century, and this trend may not continue in the future, thus qualifying the Gran Canaria blue chaffinch as an endangered or critically endangered habitat specialist.

Breeding habitat selection

Habitat preferences for successful breeding of the Gran Canaria blue chaffinch are similar to those measured in its sibling species from the nearby Tenerife island, although Fringilla polatzeki shows a remarkably lower altitudinal range and a higher preference for mature pine stands. Fringilla teydea ranges from 1,000 to 2,060 m a.s.l., reaching in the 1,500–2,000 m belt an average abundance 3.4 times higher than that recorded at 1,000–1,500 m (Carrascal & Palomino, 2005). The BCT model for F. polatzeki in Inagua shows a steep increase of habitat suitability with altitude up to 1,300 m where it stabilizes, a limit that can be understood considering that only 15.8% of Inagua is >1,300 m a.s.l. and 0.28% above 1,500 m. Thus, Inagua imposes an altitudinal restriction to F. polatzeki based on orography, but the 1,300 m a.s.l. threshold is not a true biological limit as the data of the recently established small population in La Cumbre demonstrates. The species is able to dwell at higher altitudes in this area (Delgado et al., 2016), and has shown a formidable increase in the number of breeding pairs from two in 2010 to 16 in 2016 (Rodríguez, 2016). Therefore, the altitudinal range of Gran Canaria probably imposes, per se, restrictions to the distribution of the blue chaffinch, assuming that F. teydea and F. polatzeki share similar abiotic environmental preferences as sibling species.

As for forest structure, the highest habitat suitability for the successful breeding of F. polatzeki is attained in woodland stands with more than 21 m of pine height and tree cover between 35%–55%. Practical recommendations can be derived from these results for managing the dense and relatively young pine plantations located above 1,300 m a.s.l. in other areas of Gran Canaria island (La Cumbre, Los Marteles, Moriscos-Galdar). The positive influence of pine height on habitat preferences has been also observed in F. teydea (see Carrascal & Palomino, 2005 at a broad scale, and García-del Rey et al., 2009 at the habitat use level), while the species in Tenerife island is ca. three times more abundant in thinned (53% tree cover) than in unmanaged (86%) reafforestations (García-del Rey, Otto & Fernández-Palacios, 2010). Nevertheless, the most remarkable difference between the habitat preferences of the two taxa is the ability of F. teydea to occupy young pine forests during the breeding season (e.g., Carrascal, Tellería & Valido, 1992; García-del Rey & Cresswell, 2005; García-del Rey, Otto & Fernández-Palacios, 2010), even the non-native Pinus radiata plantations (Carrascal, 1987), with densities ranging from 25 to 170 birds/km2 in woodlands with pine height ranging from 7 to 15 m. Again, the preference for well-developed and open forests of F. polatzeki in Inagua may be the consequence of the maturity of the pine forest in this area. This idea is supported by the fact that F. polatzeki is able to thrive at higher altitudes in the less mature pine forests of La Cumbre, with a survival and reproductive success very similar to that recorded in Inagua (Rodríguez & Moreno, 2008; Delgado et al., 2016).

Habitat suitability outside the main distribution area

The favourable environmental conditions for the blue chaffinch identified in Inagua suggest other natural and historic Gran Canaria pine forests that are not suitable for the species, and should be discarded in the population management plans (i.e., habitat management-restoration or translocations of individuals). This is clearly the case of Tauro and Pilancones forests, for which the predicted very low habitat suitability maps (see Fig. S3 and Fig. 4) reinforces the lack of the species throughout the historical distribution of the species in Gran Canaria island (Martín & Lorenzo, 2001). On the other hand, Tamadaba forest has more favourable habitat for the species, especially in the upper part of the two main ridges. The existence of suitable habitat for the reproduction of the species agrees with the recorded historical presence in this area, although always in low numbers up to 1991 (Moreno & Rodríguez, 2007), and recent eventual sightings since 2010 (Pascual Calabuig & Felipe Rodríguez, pers. comm., 2011 and 2016). Nevertheless, the antique photos available for the Tamadaba pine forests in the middle of the 20th century (little vegetation cover of a relatively young pine forest; http://www.fotosantiguascanarias.org/), suggest that the species was not abundant in the past. The low amount of highly suitable habitat for the blue chaffinch in Tamadaba means that this area could foster a smaller population than Inagua (see woodland area with habitat suitability >0.7 in Fig. 4; 6.58 km2 in Inagua for a population of ca. 280 individuals vs. 1.95 km2 in Tamadaba). The potential area could be further reduced considering the fragmentation of highly suitable woodland patches (see Fig. 4 and Fig. S2). This is a concern as woodland specialists usually require large patches of continuous well-preserved forests (e.g., Santos, Tellería & Carbonell, 2002; Fahrig, 2003; Devictor, Julliard & Jiguet, 2008), and habitat fragmentation negatively affects the abundance and suitability of an area for birds (e.g., Basile et al., 2016). Nonetheless, Tamadaba should be considered as a potential area for translocations of blue chaffinches, especially those sectors located at higher altitudes, with tall pine trees and higher summer rainfall. Even if in low numbers, this area would add to the two current distribution areas of the species in Gran Canaria.

Our approach has several limitations regarding other potentially important habitat and environmental variables that may influence the presence/absence of the blue chaffinch, such as the occurrence of predators (e.g., cats; Moreno & Rodríguez, 2007) or human pressure from recreation and leisure activities in some of the study areas (e.g., the central part of Tamadaba). Moreover, our study is centered on the breeding season and we do not analyze other limiting factors that may affect the blue chaffinch distribution during the wintering season.

Conclusions

Given the preference of this species for mature pine forests that are suffering forest dieback as a consequence of climate change (Martín et al., 2015), we may be witnessing the vanishing existence of an endemic woodland bird species in the eastern limit of the Canary forests. Nevertheless, the reintroduction of the species in other suitable pine forests (especially if they are located at higher altitudes), and forest management practices directed to reduce woodland fragmentation and modify habitat structure according to blue chaffinch habitat preferences, may ameliorate or counteract this vanishing trend. Recommendations for the conservation of blue chaffinch in Gran Canaria include the management of pine forests above 1,100 m a.s.l. with a summer precipitation of 13–24 mm, by reducing the cover of the canopy layer to 25–50% with the removal of some pines lower than 15 m in height. Our results demonstrate that habitat suitability obtained from modelling the location of successful breeding attempts is a good surrogate of the observed local abundance. Thus, habitat suitability can be used for the identification of potential areas for translocations of blue chaffinches or as a cost-effective tool to provide tentative population estimates.

Supplemental Information

Supplemental Information 1 Supplementary material

Tables S1 and Figures S1, S2 and S3.

Click here for additional data file.

Our acknowledgments to Cartográfica de Canarias, S.A. (GRAFCAN, http://www.grafcan.es) who did provide the cartographic information available in the Canarian Spatial Data Infrastructure (http://www.idecanarias.es/). The solar radiation information available at http://www.idecan.es belongs to the Instituto Tecnólogico de Canarias Foundation and Dobon‘s Technology, SL. We are very grateful to Joachim Hellmich, Pascual Calabuig, Ruth de Oñate and Felipe Rodríguez for the help provided in carrying out this work. Claire Jasinski and Ana Rey improved the English of the manuscript.

Additional Information and Declarations

Competing Interests

Author Contributions

Animal Ethics

Data Availability

The authors declare there are no competing interests.

Luis M. Carrascal conceived and designed the experiments, performed the experiments, analyzed the data, contributed reagents/materials/analysis tools, wrote the paper, prepared figures and/or tables, reviewed drafts of the paper.

Ángel C. Moreno performed the experiments, analyzed the data, contributed reagents/materials/analysis tools, prepared figures and/or tables, reviewed drafts of the paper.

Alejandro Delgado, Víctor Suárez and Domingo Trujillo contributed reagents/materials/analysis tools, reviewed drafts of the paper.

The following information was supplied relating to ethical approvals (i.e., approving body and any reference numbers):

The Consejería de Medio Ambiente del Cabildo de Gran Canaria gave access to carry out all the field work.

The following information was supplied regarding data availability:

We are working with an endangered species, therefore we cannot provide the locations of the nests or the trails within the Reserve where the species is more abundant for publication.

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
