# Peer review of "Habitat suitability—density relationship in an endangered woodland species: the case of the Blue Chaffinch (Fringilla polatzeki)"

_PeerJ, doi:10.7717/peerj.3771_

## Round 0.1 · original submission · Major Revisions

As you can see all three of my reviewers have raised substantial issues with this manuscript. Please respond to them.

Among many other things, they point to your need to improve the English. Seeking help from a native speaker would help, but so too would be grammar software: we use Grammarly in my lab and, even after 300 papers, I find it helps my writing.

Finally, I'm pleased you've sent us this paper on this endangered species. I'm glad to see colleagues thinking of PeerJ as a place to publish conservation studies (and I hope to see this bird one day!)

·

Basic reporting

I commend the authors on an interesting study addressing the conservation needs of an imperiled species. I found the work to be thorough and analytically rigorous. I found the graphical and tabular output was informative and well described by the headings or legends. I appreciated the authors' submission of data and code; I perused both but did not attempt to replicate the analyses. I spent most of my effort on the submitted manuscript. I did have difficulties at times understanding various prose. I provide an edited version of the submitted document attempting to provide some suggested clarity. I also note a number of instances in which the sentence structure is awkward and unclear; unfortunately, in these few cases I could not clarify the authors' meaning.

Experimental design

The research question is well defined and certainly on a topic worthy of attention. The methods are sufficiently well described, particularly given the submission of associated code. However, I do note one gap in the Methods; the specific method for estimating population size from the suitability-abundance relationship is not exactly specified in the Methods. Further, it remains a bit unclear how detectability was incorporated in this calculation. This matter requires clarification.

Validity of the findings

Aside from the reservation noted above, I have no major concerns about the research results. I believe their results derive from sound methodologies. I believe the conclusions are well supported.

Additional comments

Page 2, line 24. In my suggested edits to the submitted manuscript, I replaced nearly all instances of 'census' and replaced it with 'survey'. A census is a complete count of a population or species. The methods used in this study could not provide a census, which is exactly why suitability of habitat and its relationship to species abundance was used to infer population size.

Page 9, lines 198-199. These results for detection are best placed in the Results rather than described in the Methods.

Page 16, lines 383-384. The specific method for estimating population size from the suitability-abundance relationship is not exactly specified in the Methods. Further, it remains a bit unclear how detectability was incorporated in this calculation. It is for this reason and matters of language that I marked the recommendation for this submission as Major Revision; I actually think the revision should be fairly straight forward.

Page 19, re: resilience. It may not be the case that the small population size persisting over a period of years indicates resilience. It could just be that the species is lucky. Extinction risk is the integration of population size, trend, and variability in that trend. For this species, it could very well be that the stochastic variability of its environment has been rather benign of late; that may not continue.

Reviewer 2 ·

Basic reporting

No comment.

Experimental design

No comment.

Validity of the findings

How the model was evaluated? Did you use test data? It seems that you evaluated the model using the training data, which is not the proper way to do the model evaluation. Are those estimates (sensitivity, specificity, AUC and so on) presented in Lines 283-285 from train or test data? Without an independent test data, your model is overfitted, which means that your model may fail to predict occurrence data that is obtained independently. Please, clarify this question and, if necessary, perform a proper evaluation test.

Additional comments

The manuscript “Habitat suitability – density link in an endangered woodland species: the case of the Blue Chaffinch (Fringilla polatzeki)” by Luis M. Carrascal and collaborators presents interesting results for an endangered bird species. Specifically, the authors sampled occurrence and abundance data compiled over many years of fieldwork and they built a distribution model to test if abundance is related with environmental suitability derived from a SDM. The results have obvious applications for the species conservation and, therefore, it is desirable that such information is made public. Said that, I have some concerns that should be addressed by the authors before it is been accepted for publication.

Major issues
1. You used the same dataset to model the species occurrence and to relate abundance with environmental suitability, that is, abundance data was used to derive the presence data. I wonder if that could create some sort of overfit. Is there no record of this species outside your dataset that could be used in the modeling?

2. How the model was evaluated? Did you use test data? It seems that you evaluated the model using the training data, which is not the proper way to do the model evaluation. Are those estimates (sensitivity, specificity, AUC and so on) presented in Lines 283-285 from train or test data? Without an independent test data, your model is overfitted, which means that your model may fail to predict occurrence data that is obtained independently. Please, clarify this question and, if necessary, perform a proper evaluation test.

3. Some statistical results are worth mentioning, as follows: (i) Moran’s I value from the residuals of the regression model without spatial filters; (ii) Moran’s I value from the residuals of the regression model with spatial filters; and (iii) R2 from the relationship between abundance and suitability without spatial filters. I also suggest including the correlograms from these analyses as a Supplementary Material.

Minor issues:
Throughout the manuscript you use km2 and ha to refer to area. Please, choose one and be consistent along the whole manuscript.

Lines 14-17. I suggest you make this paragraph more general. For instance, instead of “reducing the risks associated with their presence in only one protected locality”, you could say “reducing the risks associated with species that are recorded in only one protected locality”. I also suggest you insert here the goals of the manuscript. They are not clear here.

Lines 18-24. Be more specific here. Give information such as type of algorithm used for SDM, which analysis you used to test the relationship between local abundance and suitability. In fact, you did not say explicitly that you tested this relationship.

Lines 130-143. I assume these variables were used in the modeling, right? If so, please be explicit. Furthermore, I counted 10 variables and not 12 as said in Table 2 and in Line 255, but I may have miscounted. Anyway, I suggest you rephrase this part and enumerate all variables used in the modeling to avoid such confusion and provide the variable sources in the same paragraph.

Line 206. Change “gave permisión” to the standard English language.

Line 259. Which neighborhood method did you use to select the nearest sixteen cells? Or how were they placed around the focal cell?

Line 350. Please change “favourability” to “suitability”. The former term is used when considering statistical models (e.g. logistic regression) and species prevalence (see Real et al., 2006 - DOI 10.1007/s10651-005-0003-3). Suitability is a more adequate term in this case.

Lines 398-409. Here you present number of individuals for different species in different areas, which is fine. However, for comparison reasons, I think it would be better if you present the density values (number of individuals/area) instead of that information separately. That way, data are directly comparable.

Figure 1. Insert the country names in the upper right map for the reader reference.

Figure 3. Insert the forest names in the figure.

Table 2. Rank the variables in order of importance.

Reviewer 3 ·

Basic reporting

no comment

Experimental design

no comment

Validity of the findings

no comment

Additional comments

This manuscript addresses the important issue of habitat suitability of an endemic, highly localized and vulnerable species, Fringilla polatzeki. It is noticeable that this study involved a great effort, in the field with many years of data collection but also in data analysis and interpretation. I believe it is an interesting study that can give a contribution to help in the conservation of the Blue Chaffinch and management of its main habitats. However, I have some questions and comments that I list below.

The paper is well written and, regarding manuscript structure, I think it is well organized with clear divisions with the main questions being analyzed, both in methods, results and discussion sections.

Abstract
In the results section of the abstract I miss an indication of the variables with more influence to the species.

Introduction
Regarding the introduction, the authors give a relatively broad idea of the importance of the theme being investigated, and the objectives of the study are well identified at the end.

Line 67: I miss some references regarding the statement about protected areas.

Materials and Methods
Methods section is very detailed and well organized.

Line 121: I didn’t understand what you mean with Natural Reserve surroundings.
Line 128: In Figure 1 legend it would be important to clear indicate what are the green limits, is it altitude? And can you add the limits of Inagua Integral Natural Reserve?
Line 129: Although I understand that Figures S1, S2 and S3 add detail in the information about the study areas, I don’t think it is adequate to mention them here since they are suitability maps produced as a result of your analysis.
Line 181: I compliment you for the use of distance sampling methods.
Line 252: Why did choose to repeat the BCT models 20 times?
Line 255: I think it is important for the reader to have a description of the predictor variables in the methods, so I suggest you add here a reference to Table 2.
Lines 258 and 266: Please uniform: ”one-hundred 229 m” and “100 transects units of 229 m”.

Results
Lines 282 and 415: I think it would be better if you choose “breeding” or “reproductive” and use the same in both titles.
Line 290: It would be important to have here information not just about the relative importance of the environmental variables, which is well indicated in Table 2, but also if they have a positive or negative effect (or favourable ranges).
Lines 299 and 304: I understand the interpretation of habitat suitability values, but I think it could be more clear if you add “low” and “high” when you mention 0.029 and 0.827.
Lines 322-326: I think this paragraph fits better in the discussion section.

Discussion
Overall, I enjoyed reading the Discussion, but I expected to see a broader debate about the implications of our results to the conservation of Blue Chaffinch and to the management of pine habitats in Gran Canaria.

For example, although it was not the focus of your study, do you think that factors other than habitat and environmental variables can be influencing the presence/absence of the species, as predators? Or, for example, higher human pressure from recreation and leisure activities in some of your study areas?

Do you have any data for the wintering season? How do you expect your results to differ between seasons or the areas used by Blue Chaffinch to differ?

What were the limitations found in the modelling procedure? Do you think it can be useful and applied to other species?

Finally, I think that your study is an important contribution to the conservation of Blue Chaffinch but it also gives useful indications about what would be adequate in terms of habitat management. So, based on your results, and having in account Blue Chaffinch preferences, can you present clear recommendations for the management and conservation of pine forests in Gran Canary?

Small typos mistakes:
line 18: please change study to studied
line 19: please state the scientific name of the pines
line 26: “suitability”
line 84: “International”
line 153: please remove “it”
line 191: “sampled”
line 206: “permission”
line 369: “suitability”
line 379: F. teydea should be in italics

I hope the above comments are helpful to improve this important work.

---

## Round 0.2 · Minor Revisions

You will see that most of the changes are very minor and should not take you long. I can't simply accept the paper in its current form, since it would then to press without changes. So, please address these minor changes and get it back to me quickly.

Thanks you for submitting your work to PeerJ.

Reviewer 2 ·

Basic reporting

No comment.

Experimental design

No comment.

Validity of the findings

No comment.

Additional comments

I think the updated version is much better than the previous one. The authors did a good job incorporating all reviewers suggestions when adequate. I only have one comment in the authors' conclusion. I appreciated the management practices suggested by the authors that can be adopted to preserve the species.

However, I see a problem in the following statement "Recommendations for the conservation of blue chaffinch in Gran Canaria include the management of pine forests [...] by reducing the cover of the canopy layer to 25-50% with the removal of the pines lower than 15 m in height" (lines 582-586). First, since the species is associated with pine height (lines 346-347), if you remove pines that are today lower than 15 m, you are not interrupting the succession process and avoiding that tree cover >15m be larger in the future? Such removal would not decrease habitat suitability for the species in the future? Furthermore, I think this management suggestion too species-centered. How the ecosystem or other non-bird communities would respond to the "removal of the pines lower than 15 m in height"? Probably, not well. I suggest you rephrase this sentence bearing this in mind.

Lastly, please update the reference Weber et al. (2016) to Weber et al. (2017).

Reviewer 3 ·

Basic reporting

"no comment"

Experimental design

"no comment"

Validity of the findings

"no comment"

Additional comments

The authors have significantly improved the paper, namely regarding the English spelling, and made all the changes in the manuscript according to the concerns and questions that I have raised before.
It is now clearer not only in the methods but also regarding results and management implications of this relevant study for the conservation of Fringilla polatzeki. Therefore, I recommend that it is accepted for publication.
I just suggest an overall final read to look for some spelling and typos errors. I have found a few that I list bellow:
Manuscript
line 186: a “the” is missing before project
line 196: please change Ministery to “Ministry”
line 204: best “using the average of the three surveys”?
line 223: vehicle
line 246: please add (BCT) after boosting classification trees
line 382: “the most” favourable habitat?
Line 502: steep increase
Line 826 (Table 2 legend): of “the” 12 environmental variable “used” in boosted…

Supplementary Material
Pp4, line 10: environmental is missing the first “e”

---

## Round 0.3 · accepted · Accept

Thank you for choosing PeerJ!